# Assessment of Aquatic Ecological Health Based on the Characteristics of the Fish Community Structures of the Hun River Basin, Northeastern China

**Jun Xie [1], Caiyan Wang [1], Lu Liu [1], Yuanshuai Duan [1], Bin Huo [1] and Dapeng Li [1,2,3,4,*]**

[1] College of Fisheries, Huazhong Agricultural University, Wuhan 430070, China
[2] Hubei Provincial Engineering Laboratory for Pond Aquaculture, Wuhan 430070, China
[3] Hubei Hongshan Laboratory, Wuhan 430070, China
[4] Engineering Research Center of Green Development for Conventional Aquatic Biological Industry in the Yangtze River Economic Belt, Ministry of Education, Wuhan 430070, China
* Correspondence: ldp@mail.hzau.edu.cn

**Abstract:** Long-term ecological restoration can restore aquatic ecosystems to a certain extent and alleviate the crisis of freshwater fish biodiversity. In order to explore the fish community distribution patterns and key factors after ecological restoration and the health status of the watershed, fish and environmental data were collected from 39 sampling points in the Hun River Basin in the spring and autumn of 2021. A total of 51 fish species belonging to 11 families and 37 genera were collected during the survey, and the dominant species were *Rhynchocypris lagowskii*, *Zacco platypus*, *Carassius auratus* and *Pseudorasbora parva*. Compared with the results of past studies, the number of fish species has increased. The study found that the distribution of fish along the longitudinal gradient of the watershed showed obvious spatial differences and was divided into two groups. The results of canonical correspondence analysis (CCA) showed that agricultural land, urban land and grassland were the key factors for the spatial variation in fish communities in the Hun River Basin. The results of the F-IBI evaluation showed that the health status of the Hun River was fair or above fair, among which healthy, good, fair, poor and bad points accounted for 5.13%, 30.77%, 33.33%, 25.64% and 5.13%, respectively. The upper and middle reaches of the Hun River Basin were in better health, while the lower reaches were in poorer health, which was mainly affected by the intensity of human activities in different regions. This study will help watershed managers to make targeted restoration and protection measures for different regions.

**Keywords:** fish community structure; environmental factor; aquatic ecological health; management implication

## 1. Introduction

As an important part of the global ecosystem, rivers can not only provide food and industrial, agricultural and living water, but they can also provide one of the important channels for the material cycle of the biosphere. A river has various functions such as improving the ecological environment, regulating the climate, and maintaining biological diversity [1,2]. However, in the past few decades, with the rapid industrialization and urbanization of human society, rivers all over the world have been disturbed and damaged by human activities to varying degrees [3,4]. Due to the excessive discharge of industrial sewage, domestic sewage and farmland sewage, the destruction of the riparian zone structure, the abuse of water resources and the construction of dams, rivers have become one of the most vulnerable ecosystems [5].

At present, the basic approaches of evaluating the health status of aquatic ecosystems include the species indication method and the comprehensive multi-indicator evaluation method [6,7]. As one of the comprehensive multi-indicator evaluation methods, the index

of biological integrity (IBI) concept was first proposed by Karr in 1981 [8], and subsequently, this concept framework has been constantly improved and widely applied to the evaluation of the river health using different aquatic indicative species, such as epiphytic algae [9,10], macrobenthos [7,11,12], planktons [13,14] and aquatic vascular bundle plants [15].

In terms of the health of river ecosystems, aquatic organisms are influenced not only by pollution loads, but also by habitat and hydrological conditions [16,17]. Therefore, aquatic organisms can be used to obtain valuable information about the comprehensive impacts of various pressure sources and environmental variables. Fish are one of the most commonly used communities in ecological health assessments [18]. Mamun et al. [19] believed that fish are important aquatic organisms in the aquatic ecosystem, with long life cycles, a wide distribution, obvious morphological characteristics, easy identification and strong activity abilities. They can survive in most aquatic ecosystems, and they can comprehensively reflect the information and environmental changes in the watershed aquatic ecosystem. Therefore, it was suggested that the fish biological integrity index should be selected to assess the health of the watershed aquatic ecosystem. It was found that the community structures of fish are affected by many factors, such as land use, hydrological conditions, topography, water quality and biological effects [20–23]. However, due to regional differences, fish community structures have different response sensitivities to environmental factors, and so analyzing the characteristics and influencing factors of fish community structures are of great significance to the protection of the river ecosystem [24].

As the second largest river in Liaoning Province, the Hun River flows through cities such as Fushun, Shenyang, Liaoyang and Anshan, and it provides important ecological service functions for the development of local social economy. However, with the rapid development of industry, agriculture and cities, the ecosystem of the Hun River Basin has been severely damaged, and the biodiversity has declined severely [25]. Comprehensive monitoring of its fish community structures and the identification of key factors driving changes in its fish communities are critical for establishing and evaluating management strategies [26,27]. In order to effectively protect and restore the ecosystem of the Hun River Basin, the state and local governments have taken a series of measures since the implementation of the national water pollution control and treatment major science and technology project (e.g., water pollution control, water pollution treatment, afforestation and seasonal fishing bans). However, the results of fish community structures and health assessments in the Hun River Basin have primarily been based on data from the period 2010 to 2012 [28–30], and these investigations were limited to timely assessments and the improvement of management strategies. At present, the national water pollution control and treatment major science and technology project has ended, and the current status of the Hun River Basin after treatment and restoration is unknown.

The purpose of this study was based on the following fish data and environmental factors: (1) to analyze the structures of the fish communities in the Hun River Basin after ecological restoration, and to clarify the key factors affecting the distribution of these fish communities; (2) to develop a feasible fish integrity index and evaluate the health status of the aquatic ecosystem of the Hun River Basin; and (3) to propose corresponding restoration and protection measures for different areas.

## 2. Materials and Methods

### 2.1. Study Area and Sampling Site

The Hun River Basin (40°20′–41°00′ N, 122°20′–125°20′ E) is located in the central and eastern part of Liaoning Province, with a total length of 415 km and a basin area of $1.14 \times 10^4$ km². The annual rainfall is 404–934 mm, and most of the precipitation occurs during the flood season from June to September. The source of the Hun River originates from the high elevations area of Changbai Mountain, with less human disturbance and high forest coverage, and it flows through important cities such as Fushun, Shenyang, Liaoyang and Anshan. Along the middle and lower reaches, the land-use types are gradually dominated by agriculture, industry and urban areas. There is a large reservoir in the

middle and upper reaches—the Dahuofang Reservoir—which can provide water resources for life, industry and agriculture. A clear environmental gradient can be observed in the Hun River Basin. The area above the Dahuofang Reservoir is a mountainous-hilly area with high forest coverage and weak human disturbance, and the downstream area is a plain area with dense industrial areas, a high degree of land development and serious human disturbance.

A total of 39 sampling points were selected in this study in the spring and autumn of 2021, including 18 sites in the river's main stream (H01–H18) and 21 sites in the tributary (H19–H39) (Figure 1). The sampling sites were selected in the field considering the representativeness and accessibility of the habitat.

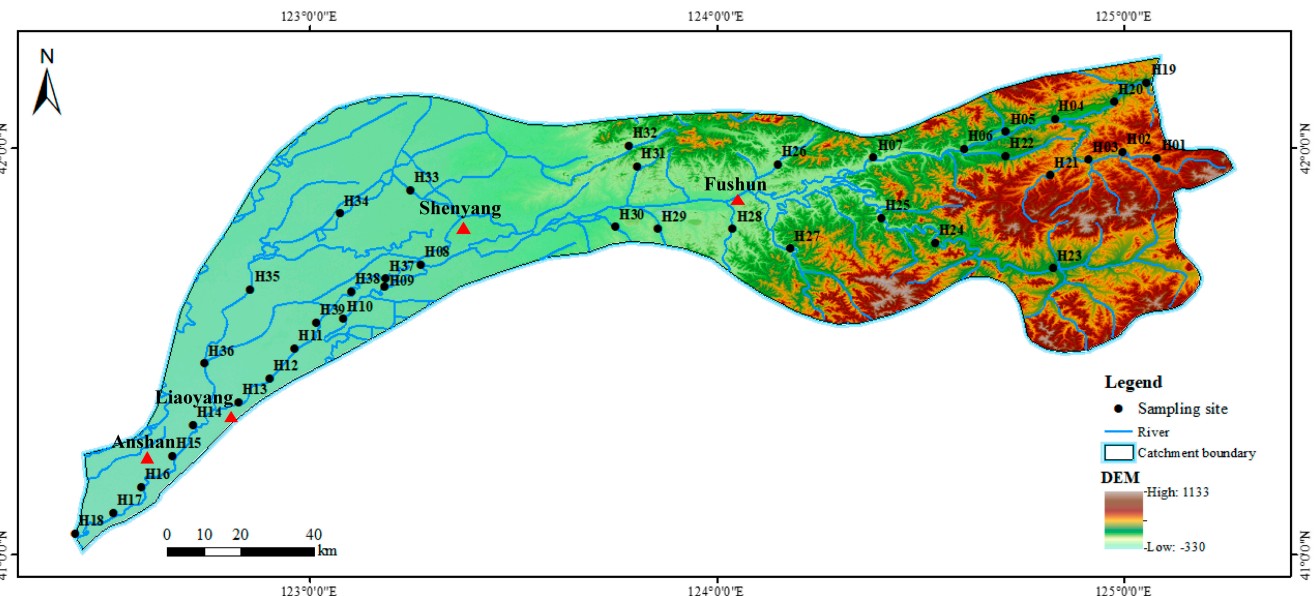

**Figure 1.** Sampling sites in the Hun River Basin, China. A digital elevation model (DEM) is shown, with low to high elevation as indicated in the legend.

### 2.2. Data Collection

#### 2.2.1. Fish Samples

Fish samples were collected using electrofishing and gill nets. In the river section with a water depth of less than 1.5 m, electrofishing was used to collect fish within 200 m upstream and downstream of the sampling point, and the sampling time was 30 min. In river sections with water depths of greater than 1.5 m, fish were collected with gill nets (mesh size: 1 cm × 1 cm; 2 cm × 2 cm; and 5 cm × 5 cm), and the time period for hanging the nets was approximately 2 h [31]. The captured fish were identified on-site, and some basic biological data was obtained before they were released back into the river, which was intended to reduce habitat disturbance. The fish that could not be identified in the field were photographed and stored in 10% formalin, and then they were brought back to the laboratory for identification [32].

#### 2.2.2. Environmental Factors

Water sample collection and sample storage were carried out in accordance with the Chinese national experimental standards [33]. The water samples were collected with a plastic bottle (500 mL) at each sampling point, and we performed acidification (by sulfuric acid, $H_2SO_4$), placed them in a cryogenic freezer (to minimize the deterioration before chemical analysis) and transported to a laboratory for chemical analysis. The total nitrogen (TN, mg/L), ammonia nitrogen ($NH_3$-N, mg/L), nitrate nitrogen ($NO_3$-N, mg/L), nitrite nitrogen ($NO_2$-N, mg/L), total phosphorus (TP, mg/L), soluble reactive phosphorus (SRP, mg/L) and permanganate indexes ($COD_{Mn}$, mg/L) were determined

according to the Chinese national standard method for water and wastewater detection (standard methods for the examination of water and wastewater, 2002). Concurrently, other parameters, including pH, dissolved oxygen (DO, mg/L) and water temperature (WT, °C), were measured on site by a multi-parameter water quality analyzer (HQ40D, USA), and the flow velocity (FV, m/s) was determined using a direct reading flow velocity meter (FP111), the transparency (SD, m) was measured through a Seclhi disk and the altitude (Alt, m), longitude and latitude were measured with a portable UniStrong G510.

Agricultural land (AL, %), forest land (FL, %), grassland (GL, %) and urban land (UL, %) have been reported to have significant impacts on fish communities [34,35], and so land-use data for each sampling point were extracted on the GlobeLand30 map using ArcGIS 10.2 [36], and then we analyzed the relationship between the landscapes and the fish communities. The watershed boundaries were determined through the HydroSHEDS database (https://www.hydrosheds.org/ (accessed on 8 May 2022)). The watershed range extracted in this study was not a complete watershed, but rather, it was a small-scale watershed boundary covering all sampling points. The accuracy of the river network was based on BasinATLAS_v10_lev12 data [37]. The land-use data was from 2020.

### 2.3. Construction of the F-IBI

#### 2.3.1. Selection of Reference Sites

Since there were almost no sites for human interference, most past studies have referred to optimal physical, chemical and biological conditions to determine the least-disturbed site as a reference site [17,38,39]. Comprehensively considering the field investigation conditions of the Hun River Basin, the relatively undamaged sampling sites were selected as the reference sites. The standards used for reference points were: selected sample points that met China's national Class III water standards, except for the total nitrogen and Shannon–Wiener indexes ($H'$) that were greater than or equal to 2 [40–42].

#### 2.3.2. Screening of Candidate Indicators

In order to reflect the changes in the fish community structures on the environmental gradient of the Hun River Basin, this study proposed five types of candidate index systems, which included species composition and abundance, nutritional structure, tolerance, reproductive co-location group, and habitat preference (Table 1). The selected metrics for the development of the F-IBI were chosen using the following step-wise screening procedures: (1) Discrimination power test: we compared the overlap of the box IQ (interquartile ranges; 25% to 75% quantile range) of each candidate index between the reference points and the damaged points as follows: no overlap in the box range, IQ = 3; boxes partially overlapped, but each median was outside the range of the other boxes, IQ = 2; only one median was within the box range, IQ = 1; and median of both boxes was within the other box range, IQ = 0. The candidate indicators with IQs of greater than or equal to 2 were reserved for further analysis. (2) Redundancy test: a Spearman correlation analysis was carried out on the retained indicators, and we excluded one of the indicators with a greater correlation ($|r| > 0.75$) [42].

#### 2.3.3. Calculation of the F-IBI

Among the standardized methods for the evaluation of IBI systems, the ratio method is considered to be the most accurate and effective [43], and so the ratio method was used in this study. We used candidate indicators that decreased due to the disturbance response, with their 95% quantile as the best expected value, and the index value of each point was equal to the actual value of the sample point divided by the best expected value. We also used candidate indicators that increased due to disturbance response, with their 5% quantile as the best expected value. The index score was calculated as follows: (maximum—actual value)/(maximum—best expected value).

We added each parameter value to the F-IBI value, and 95% of the division of the F-IBI value distribution of all sampling points was used as the health threshold. Sample sites

with F-IBI values higher than this standard were evaluated as healthy. F-IBI values below this standard were divided into four intervals from high to low, corresponding to good, fair, poor and bad ecosystem health.

**Table 1.** Candidate indicators for the F-IBI and their expected directions of response to disturbance.

| Indicator Type | Candidate Indicators | Response to Disturbance |
|---|---|---|
| Species composition and abundance | Number of individuals (M1) | Decrease |
| | Species of fish (M2) | Decrease |
| | Percentage of Cypriniformes (M3) | Decrease |
| | Percentage of Perciformes (M4) | Increase |
| | Percentage of Cyprinidae (M5) | Increase |
| | Percentage of Cobitidae (M6) | Decrease |
| | Percentage of Gobiidae (M7) | Increase |
| | Shannon–Wiener index (M8) | Decrease |
| | Margalef index (M9) | Decrease |
| Nutritional structure | Percentage of carnivorous fish (M10) | Decrease |
| | Percentage of herbivorous fish (M11) | Decrease |
| | Percentage of omnivorous fish (M12) | Increase |
| Tolerance | Percentage of tolerant fish (M13) | Increase |
| | Percentage of sensitive fish (M14) | Decrease |
| Reproductive co-location group | Percentage of floating-egg fish (M15) | Decrease |
| | Percentage of sticky-egg fish (M16) | Increase |
| | Percentage of sinking-egg fish (M17) | Decrease |
| | Percentage of fish with special spawning types (M18) | Increase |
| Habitat preference | Percentage of pelagic fish (M19) | Decrease |
| | Percentage of middle and lower layers (M20) | Increase |
| | Percentage of bottom fish (M21) | Decrease |

### 2.4. Data Analysis

Fish community diversity was analyzed using the Shannon–Weiner index ($H'$), Margalef index ($D$), Pielou index ($J$) [44] and relative importance index ($IRI$) [45]. The above indicators were calculated with the following formula:

$$H = -\sum_{i=1}^{S} P_i ln P_i \tag{1}$$

$$D = (S - 1)/lnN \tag{2}$$

$$J = H'/lnS \tag{3}$$

$$IRI = (P_i + W_i) \times F_i \tag{4}$$

where $S$ and $N$ are the total number of fish species and individuals in the sample site, respectively, $P_i$ is the proportion of the number of fish species $i$ in the total number of individuals, $W_i$ is the proportion of the weight of fish species $i$ in the total weight, $F_i$ is the frequency of the occurrence of fish species $i$, $IRI \geq 1000$ is the dominant species, $100 \leq IRI < 1000$ is the common species, $10 \leq IRI < 100$ is the general species and $IRI < 10$ is the occasional species.

Detrended correspondence analysis (DCA) was performed on the fish abundance data of each sampling point, and the study found that the gradient length of the longest axis was greater than 4, and so the typical correspondence analysis (CCA) was selected to analyze the relationship between fish community and environmental factors. Before performing CCA analysis, all data were transformed by log (x + 1), and then a Monte Carlo permutation test (MCT) was performed [46]. CCA analysis was performed using CANOCO 5.0 software.

Latent spatial groupings of fish community structures were determined using a hierarchical cluster analysis (CA) approach based on the group-average connectivity of the

Bray–Curtis similarity matrix. The non-metric multidimensional scale (NMDS) was used to further validate the accuracy of the clustering results. The analysis of similarities (ANOSIM) was used to test whether the differences between the fish community structures between the groups were statistically significant. The Mann–Whitney U nonparametric test was used to test the environmental parameters and the fish biological parameters between the different groups.

Spearman correlation analyses and Mann-Whitney U nonparametric tests were performed using SPSS 26.0 software. CA, NMDS and ANOSIM were performed using PRIMER-E (version 6).

## 3. Results

### 3.1. Composition of Fish Species

We collected a total of 12,220 individual fish from 39 sites in the Hun River Basin, with 5727 and 6493 in the spring and autumn, respectively. A total of 51 species of fish were identified in the two seasons, belonging to 7 orders, 11 families and 37 genera (Figure 2). The largest number of species was Cyprinidae (27 species, 52.94% of the total species), followed by Cobitidae (7 species, 13.73%), Gobiidae (7 species, 13.73%), Bagridae (2 species, 3.92%) and Eleotridae (2 species, 3.92%). There was only one species of Petromyzonidae, Siluridae, Osmeridae, Adrianichthyidae, Gasterosteidae and Channidae (1.96%). The dominant species of the fish communities in the Hun River Basin were *Carassius auratus*, *Zacco platypus*, *Pseudorasbora parva* and *Rhynchocypris lagowskii*, with IRI values of 4267.26, 3367.27, 1598.23 and 1183.95, respectively (Table 2). The fish with higher frequencies were *Carassius auratus*, *Zacco platypus*, *Pseudorasbora parva* and *Misgurnus anguillicaudatus*, all of which were tolerant species, and they were distributed in more than 60% of the sampling points.

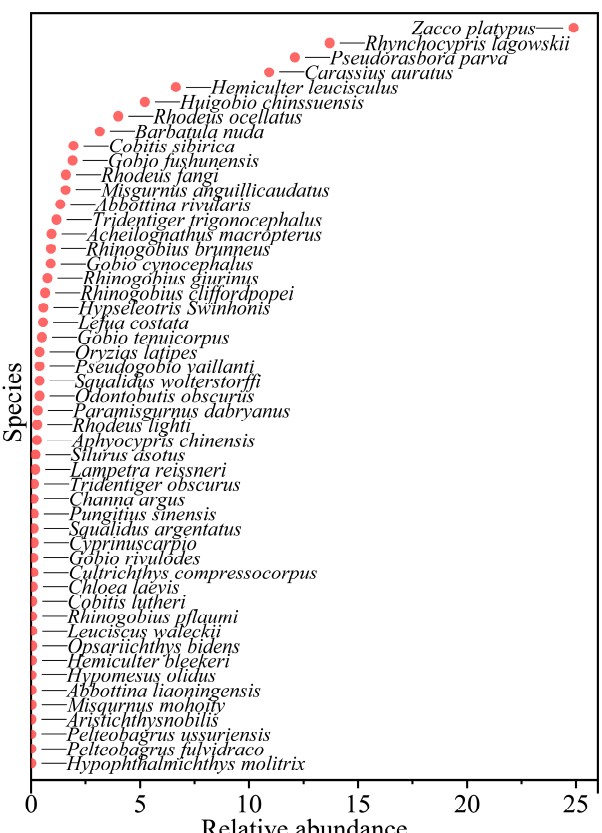

**Figure 2.** Fish species composition in the Hun River Basin.

**Table 2.** The species composition of the fish in the Hun River Basin, showing the total number of individuals (*N*), total biomass (*W*) and index of relative importance (*IRI*).

| Family | Species | *N* (ind.) | *W* (g) | *IRI* |
|---|---|---|---|---|
| Petromyzonidae | *Lampetra reissneri* | 23 | 218.57 | 6.41 |
| Cyprinidae | *Zacco platypus* | 3041 | 10,500.84 | 3367.27 |
| | *Opsariichthys bidens* | 6 | 111.02 | 3.28 |
| | *Aphyocypris chinensis* | 33 | 35.99 | 5.89 |
| | *Leuciscus waleckii* | 6 | 372.54 | 2.98 |
| | *Rhynchocypris lagowskii* | 1674 | 5822.03 | 1183.95 |
| | *Hemiculter leucisculus* | 812 | 194.97 | 544.05 |
| | *Hemiculter bleekeri* | 5 | 70.79 | 0.36 |
| | *Cultrichthys compressocorpus* | 13 | 56.12 | 3.82 |
| | *Acheilognathus macropterus* | 115 | 48.08 | 60.88 |
| | *Rhodeus ocellatus* | 488 | 189.77 | 316.43 |
| | *Rhodeus lighti* | 35 | 41.6 | 4.43 |
| | *Rhodeus fangi* | 195 | 106.34 | 79.81 |
| | *Pseudorasboraparva* | 1479 | 1239.31 | 1598.23 |
| | *Gobio fushunensis* | 232 | 857.8 | 48 |
| | *Gobio cynocephalus* | 110 | 1033.08 | 42.61 |
| | *Gobio rivulodes* | 13 | 29.53 | 0.92 |
| | *Gobio tenuicorpus* | 61 | 501.42 | 31.5 |
| | *Squalidus argentatus* | 15 | 34.59 | 2.8 |
| | *Squalidus wolterstorffi* | 48 | 88.38 | 5.33 |
| | *Huigobio chinssuensis* | 637 | 1736.69 | 416.27 |
| | *Abbottina rivularis* | 164 | 672.42 | 91.29 |
| | *Abbottina liaoningensis* | 4 | 7.3 | 0.22 |
| | *Pseudogobio vaillanti* | 48 | 221.31 | 10.89 |
| | *Cyprinus carpio* | 14 | 233 | 41.7 |
| | *Carassius auratus* | 1335 | 8360.97 | 4267.26 |
| | *Aristichthys nobilis* | 2 | 954 | 3.54 |
| | *Hypophthalmichthys molitrix* | 1 | 257.66 | 0.96 |
| Cobitidae | *Barbatula nuda* | 386 | 1887.57 | 299.93 |
| | *Lefua costata* | 68 | 249.15 | 11.68 |
| | *Cobitis sibirica* | 237 | 526.56 | 124.66 |
| | *Misgurnus anguillicaudatus* | 193 | 1083.75 | 263.78 |
| | *Misqurnus mohoity* | 3 | 9.99 | 0.3 |
| | *Cobitis lutheri* | 7 | 39.89 | 1.17 |
| | *Paramisgurnus dabryanus* | 37 | 193.29 | 41.71 |
| Bagridae | *Pelteobagrus fulvidraco* | 1 | 32.45 | 0.86 |
| | *Pelteobagrus ussuriensis* | 1 | 36.06 | 0.15 |
| Siluridae | *Silurus asotus* | 25 | 2048.52 | 96.32 |
| Osmeridae | *Hypomesus olidus* | 4 | 3.11 | 0.1 |
| Adrianichthyidae | *Oryzias latipes* | 48 | 15.13 | 12.73 |
| Gasterosteidae | *Pungitius sinensis* | 15 | 17.78 | 1.52 |
| Eleotridae | *Odontobutis obscurus* | 47 | 193.25 | 6.77 |
| | *Hypseleotris swinhonis* | 69 | 53.36 | 29.99 |
| Gobiidae | *Tridentiger trigonocephalus* | 143 | 722.04 | 11.28 |
| | *Tridentiger obscurus* | 16 | 129.64 | 0.81 |
| | *Rhinogobius cliffordpopei* | 78 | 105.11 | 40.98 |
| | *Rhinogobius brunneus* | 111 | 117.14 | 51.02 |
| | *Rhinogobius giurinus* | 91 | 97.31 | 24.88 |
| | *Rhinogobius pflaumi* | 6 | 15.48 | 0.18 |
| | *Chloea laevis* | 10 | 13.16 | 0.26 |
| Channidae | *Channa argus* | 15 | 3051.98 | 104.94 |

*3.2. Spatial Distribution Patterns of Fish Communities*

The hierarchical cluster analysis (CA) results of the group-averaged connectivity based on the Bray–Curtis similarity matrix showed that all sampling points could be spatially divided into two groups (Figure 3). The first group (Group A) was primarily

distributed in the middle and upper reaches of the Hun River Basin, which belonged to the mountainous-hilly area, with a high forest coverage rate and less human disturbance. It was distributed with sensitive species such as *Rhynchocypris lagowskii, Cobitis sibirica* and *Barbatula nuda*. The second group (Group B) was primarily distributed in the lower reaches of the Hun River, which was located in a low-altitude plain area with a dense population, developed industry and agriculture, high urbanization level and serious human disturbance, and it was primarily populated with tolerant species such as *Hemiculter leucisculus, Pseudorasbora parva, Carassius auratus, Misgurnus anguillicaudatus, Acheilognathus macropterus* and *Rhodeus ocellatus*, while the sensitive species were rarely distributed. Group A contained 21 sampling points, and its dominant species were *Zacco platypus, Rhynchocypris lagowskii, Carassius auratus, Huigobio chinssuensis* and *Pseudorasbora parva*. Group B contained 18 sampling points, and its dominant species were *Carassius auratus, Pseudorasbora parva* and *Hemiculter leucisculus*.

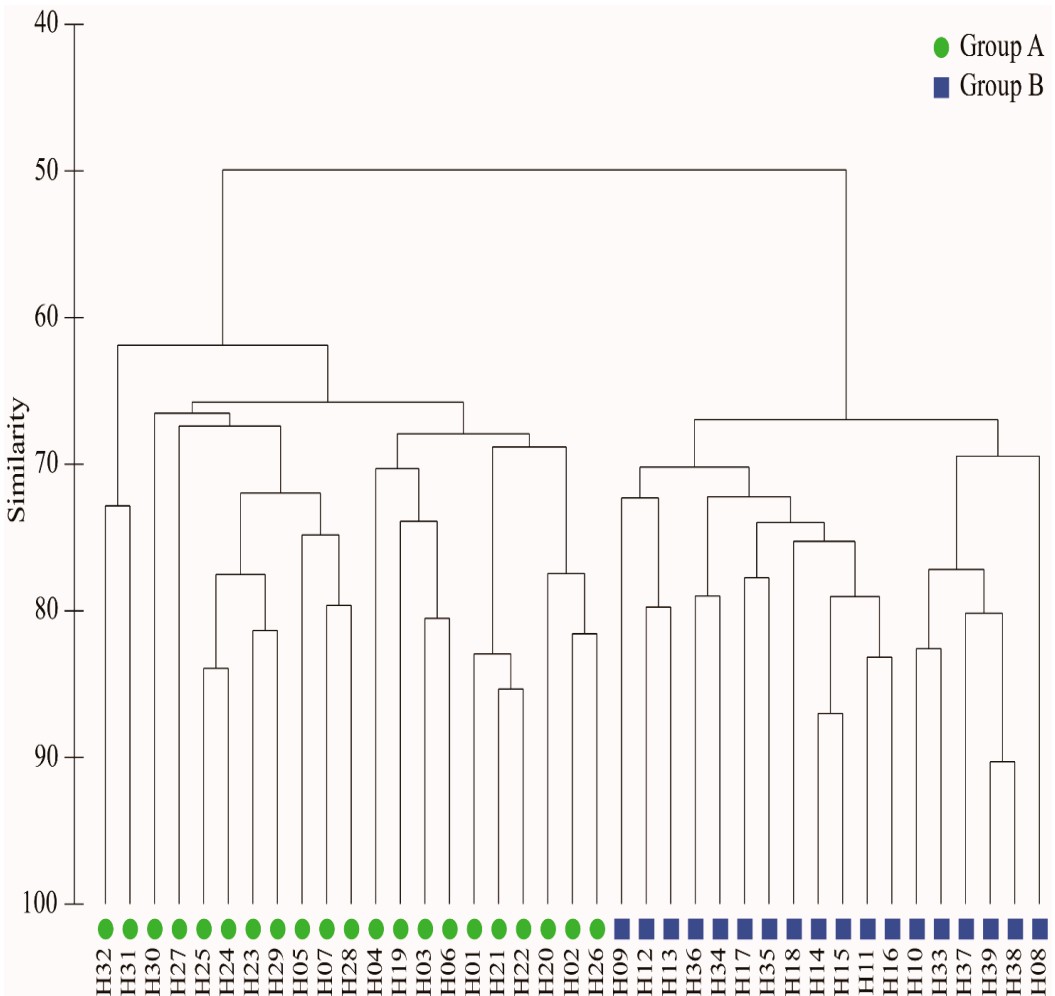

**Figure 3.** Cluster analysis of the 39 sampling sites in the Hun River Basin based on the fish species relative abundance data.

The results of the NMDS analysis showed that the two groups were significantly different on the sorting axis, and they only partially overlapped (Figure 4). The CA analysis results of the fish were consistent with the NMDS analysis results, and the grouping was meaningful. One-way ANOSIM analysis further revealed that there was a significant difference in the fish community structures among the cluster groups (R = 0.811, $p < 0.05$). The Mann–Whitney U test on the fish biological parameters between the different groups

in the Hun River Basin found that the Shannon–Wiener index and the Margalef index in Group A were higher than those in Group B (Figure 5).

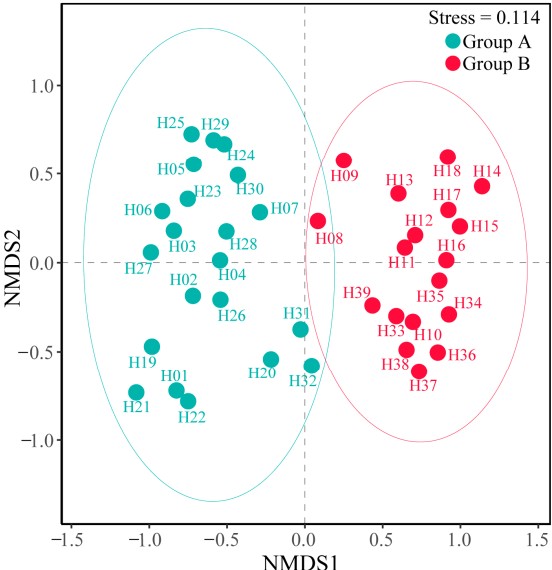

**Figure 4.** NMDS classification results of the fish communities based on the cluster analysis.

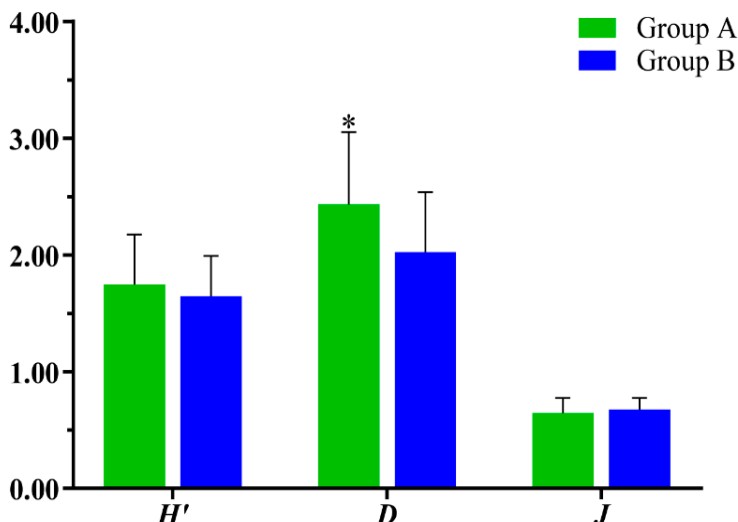

**Figure 5.** Spatial variances of the species diversity for the different groups (*, $p < 0.05$). $H'$, $D$ and $J$ represent the Shannon–Wiener index, Margalef index and Pielou's index, respectively.

### 3.3. Relationships between Fish Communities and Environmental Parameters

The results of the Mann–Whitney U nonparametric tests showed that there were significant differences in the physical and chemical factors of the water environment between the mid-upstream and downstream areas. In addition, there were significant differences in the altitude, proportion of forest land, proportion of agricultural land, proportion of grassland and proportion of urban land between the mid-upstream and downstream areas. The environmental characteristics of the mid-upstream area were primarily high altitude, high forest land and grassland, low agricultural land and urban land, and good water environment quality. The environmental characteristics of the downstream area were low altitude, low forest land and grassland, high agricultural land and urban land, and poor water environment quality (Table 3).

**Table 3.** The comparison of environmental factors between the middle-upper reaches (Group A) and downstream (Group B) in the Hun River Basin.

| Environmental Factors | Mid-Upstream | Downstream | *p* |
|---|---|---|---|
| TN (mg/L) | 5.30 ± 1.57 | 7.97 ± 1.64 | 0.000 ** |
| TP (mg/L) | 0.19 ± 0.05 | 0.32 ± 0.10 | 0.000 ** |
| NH$_3$-N (mg/L) | 0.25 ± 0.21 | 1.01 ± 0.88 | 0.000 ** |
| NO$_3$-N (mg/L) | 3.04 ± 1.07 | 4.31 ± 1.41 | 0.003 ** |
| NO$_2$-N (mg/L) | 0.04 ± 0.05 | 0.16 ± 0.05 | 0.000 ** |
| SRP (mg/L) | 0.08 ± 0.14 | 0.14 ± 0.11 | 0.000 ** |
| COD$_{Mn}$ (mg/L) | 3.52 ± 0.91 | 6.64 ± 1.53 | 0.000 ** |
| Chl-a (mg/L) | 8.12 ± 4.11 | 40.65 ± 30.75 | 0.000 ** |
| SD (m) | 0.44 ± 0.20 | 0.49 ± 0.22 | 0.621 |
| FV (m/s) | 0.56 ± 0.19 | 0.25 ± 0.21 | 0.000 ** |
| DO (mg/L) | 10.67 ± 0.97 | 10.36 ± 2.56 | 0.714 |
| pH | 8.09 ± 0.25 | 8.47 ± 0.26 | 0.000 ** |
| WT (°C) | 15.72 ± 1.61 | 17.86 ± 1.05 | 0.000 ** |
| Alt (m) | 206.48 ± 112.42 | 15.22 ± 8.53 | 0.000 ** |
| AL (%) | 27.46 ± 11.21 | 76.74 ± 11.78 | 0.000 ** |
| FL (%) | 58.62 ± 19.08 | 1.27 ± 1.41 | 0.000 ** |
| GL (%) | 8.97 ± 5.14 | 1.73 ± 1.46 | 0.000 ** |
| UL (%) | 4.94 ± 5.83 | 20.26 ± 11.12 | 0.000 ** |

Note: **, *p* < 0.01.

Based on the environmental and fish data collected in 2021, the responses of the fish communities to the environmental factors were studied using CCA. The results of the CCA analysis showed that agricultural land (*p* = 0.002), urban land (*p* = 0.006) and grass land (*p* = 0.046) were significant environmental factors affecting the structures of the fish communities in the Hun River Basin. At the same time, different species of fish had different responses to the environmental factors. Most fish were negatively correlated with AL, with the primary species of fish affected being *R. lagowskii*, *H. chinssuensis*, *B. nuda*, *C. sibirica* and *A. rivularis*. However, *A. macropterus*, *H. leucisculus*, *R. ocellatus* and *C. auratus* were positively correlated with AL. The relative abundance of *p. parva*, *O. latipes*, *R. fangi* and *C. auratus* showed a significant positive correlation with UL, and the relative abundance of *Z. platypus* and *G. fushunensis* had a significant positive correlation with GL (Figure 6a).

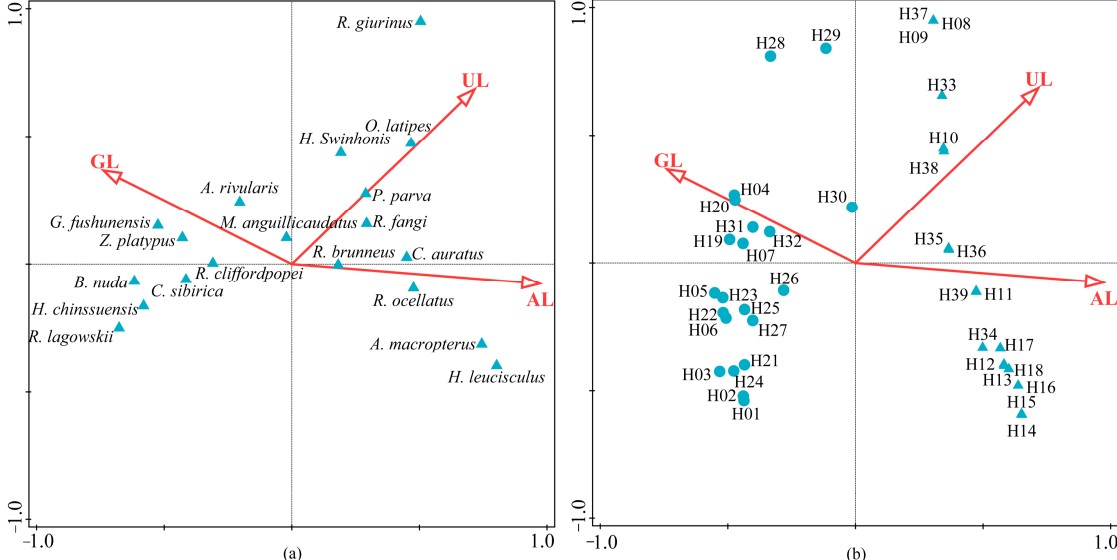

**Figure 6.** Ordination plot of the CCA for the sampled fish–environment factors (**a**) and site–environment factors (**b**) in the Hun River Basin (b: the circles represent the sites located mid-upstream, while the triangles represent the sites located downstream).

Combined with the CCA analysis diagram of the fish–environment factors and sampling site–environment factors, it could be seen that the dominant species in the downstream area were tolerant species (e.g., A. macropterus, H. leucisculus, R. ocellatus, *P. parva*, *O. latipes*, *R. fangi* and *C. auratus*) preferring habitats with high agricultural land and urban land coverage. In contrast, intolerant fish (e.g., *R. lagowskii*, *H. chinssuensis*, *B. nuda*, *C. sibirica*, *A. rivularis* and *Z. platypus*) prevailed in the middle and upper reaches, preferring habitats with high grass land, low agricultural land and urban land coverage (Figure 6).

### 3.4. Selection of Core Metrics and Establishment of the F-IBI

The results of the distribution range test showed that the candidate indicators were all suitable. The discriminative ability analysis of 21 indicators was analyzed by the box diagram method, and it was found that there were 12 candidate indicators with an IQ of greater than or equal to two, namely, M1, M2, M6, M8, M9, M13, M14, M16, M17, M19, M20 and M21 (Figure 7). Finally, a Spearman analysis was performed to select M1, M2, M6, M8, M13, M14, M16, M17, M19 and M21 as the core indicators for participation in the construction of the F-IBI (Figure 8).

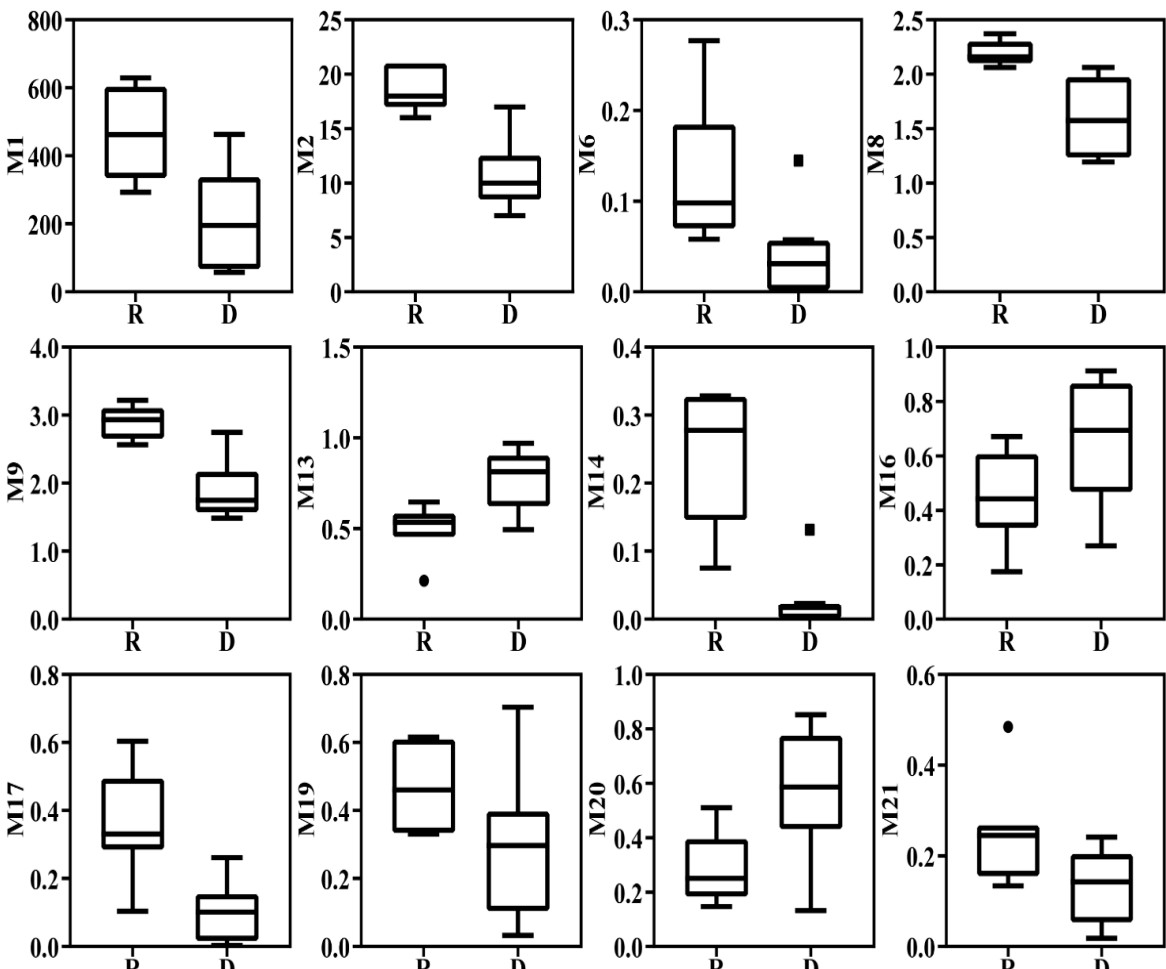

**Figure 7.** Box plots of the 12 candidate metrics between the reference sites (R) and the disturbed sites (D).

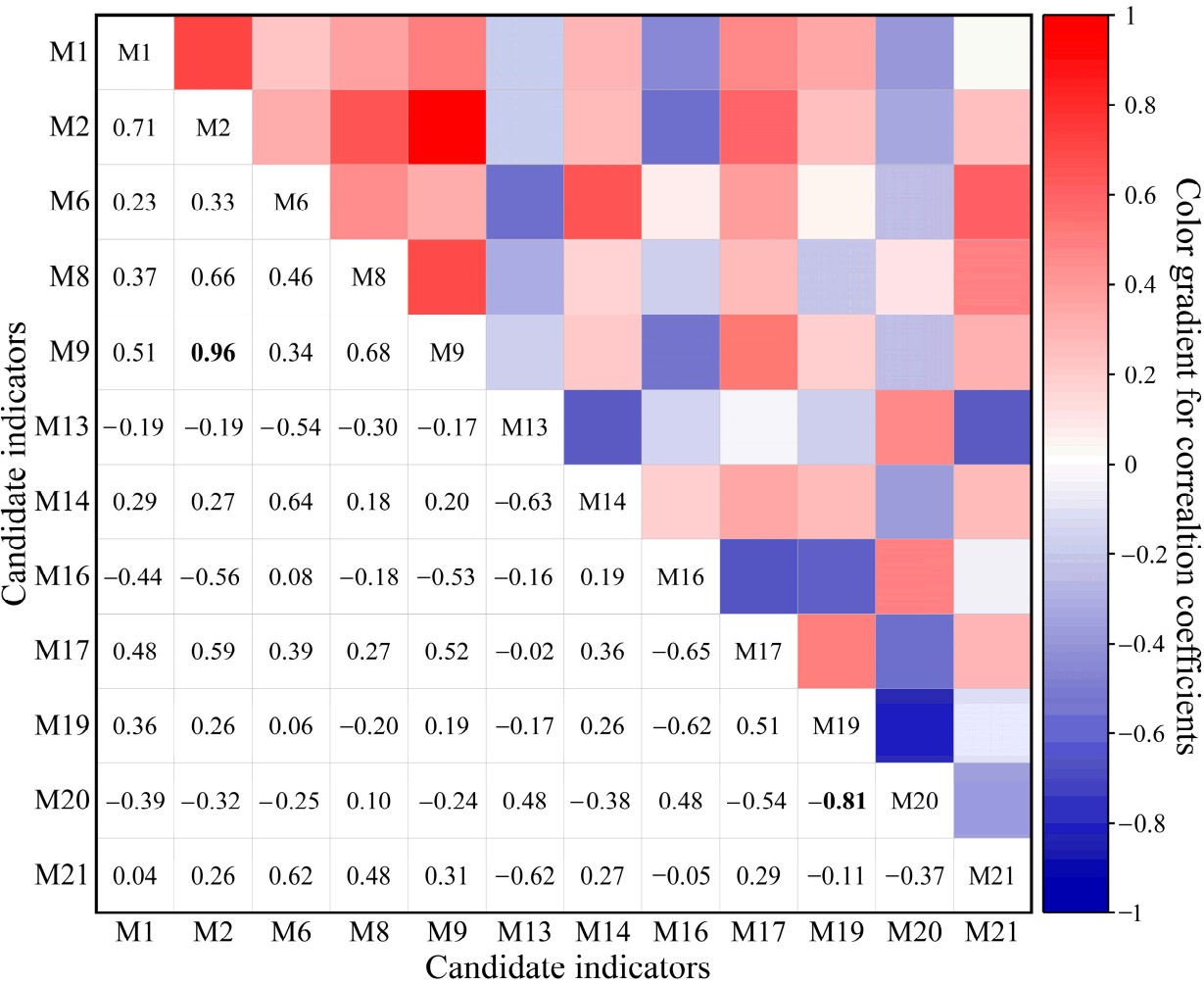

**Figure 8.** Spearman related analysis results between the 12 candidate indicators.

The health evaluation standards of the Hun River Basin were as follows: F-IBI $\geq$ 7.92, healthy; 5.94 $\leq$ F-IBI < 7.92, good; 3.96 $\leq$ F-IBI < 5.94, fair; 1.98 $\leq$ F-IBI < 3.96, poor; and F-IBI < 1.98, bad (Table 4).

**Table 4.** Criteria of health assessment of the Hun River Basin using the F-IBI.

| Health Status | Healthy | Good | Fair | Poor | Bad |
|---|---|---|---|---|---|
| F-IBI | $\geq$7.92 | 5.94–7.92 | 3.96–5.94 | 1.98–3.96 | <1.98 |

*3.5. Distribution Patterns of Aquatic Ecological Health*

According to the F-IBI evaluation criteria, in 2021, 5.13% of the sampling points were evaluated as being healthy, 30.77% as good, 33.33% as fair, 25.64% as poor and 5.13% as bad (Figure 9). Among these sampling points, the middle and upper reaches of the region had points with better evaluation results and its average F-IBI was 6.00, while the downstream area had points with poor evaluation results and its F-IBI average was 3.61. The F-IBI value for the middle and upper reaches of the Hun River Basin was significantly higher than that of the downstream, and its health status was better (Figure 10).

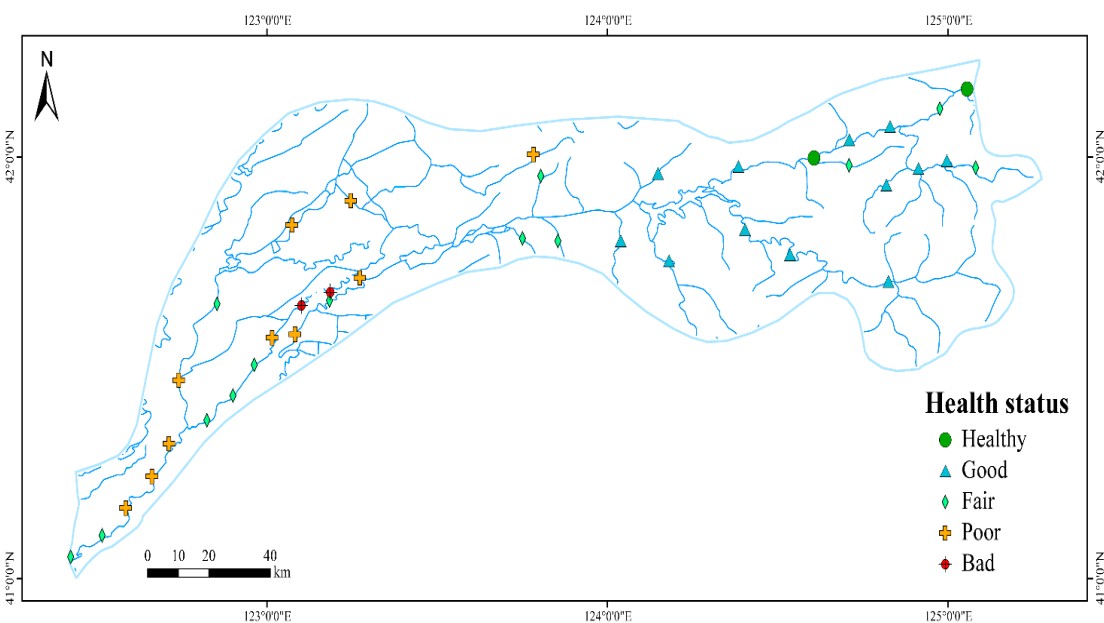

**Figure 9.** Spatial distribution based on the fish biological integrity index (F-IBI) health assessment levels.

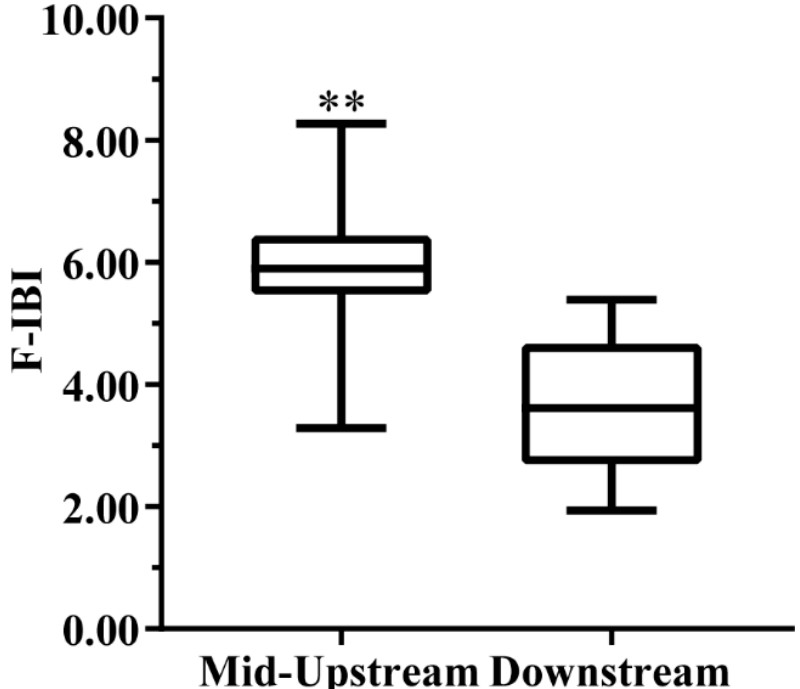

**Figure 10.** Spatial dynamic box diagram of the aquatic ecological health in the Hun River Basin (**, $p < 0.01$).

## 4. Discussion

### 4.1. Fish Community Structures and Distribution Characteristics

A total of 51 species of fish were collected in the two seasons, belonging to 7 orders, 11 families and 37 genera. The dominant species were *Carassius auratus*, *Zacco platypus*, *Pseudorasbora parva* and *Rhynchocypris lagowskii*, all of which were small fish. Compared with the number of fish species (32–34) surveyed during the period 2010 to 2012 [28–30], there was a significant increase in the number of species. It can be seen that since the implementation of the National Water Pollution Control and Treatment Major Science and

Technology Project, state and local governments have taken a series of measures (e.g., water pollution control, water pollution treatment, afforestation and seasonal fishing bans) to effectively improve the habitat environment and alleviate the freshwater fish biodiversity crisis [47,48].

The spatial distribution pattern of organisms is the adaptive distribution characteristics of species formed under the long-term combined action of heterogeneous habitats and human disturbances [49]. This study found that on the vertical gradient (from top to bottom) in the Hun River Basin, with the gradual decline in altitude, the distribution of fish showed obvious spatial differences. The fish community was divided into two groups: Group A was primarily distributed in the middle and upper reaches of the Hun River, and it was located in a plateau hilly area with high vegetation coverage and weak human disturbance, while Group B was primarily distributed in the lower reaches of the Hun River, and it was located in a low-altitude plain area with a dense population, developed industry and agriculture, a high level of urbanization, and serious human disturbance. Therefore, the fish richness, Shannon–Wiener diversity index, Margalef richness index and the percentage of sensitive species in the middle and upper reaches of the Hun River Basin were higher than those in the lower reaches. This distribution pattern is derived from the response of fish to natural environment and human disturbances, and it reflects the selection and adaptation of fish to the differences in natural geographical environment and human disturbance [50]. Compared with the three fish groups reported in the past ten years [25,29], this study was only divided into two fish groups. The reason for this change in distribution pattern may be the environmental homogeneity between the upstream and midstream regions.

### 4.2. Response of Fish Communities to Environmental Factors

The spatiotemporal variation in fish community structures is primarily caused by the heterogeneity of environmental factors on the spatial and temporal scales, and the primary factors affecting community structures include the natural environment, land use and water environment quality, etc. [24,51–53]. From the perspective of the spatial distribution pattern of the whole watershed, there were significant differences in the environmental factors between the middle and upper reaches and lower reaches. Canonical correspondence analysis showed that agricultural land, urban land and grass land were the key factors forming the spatial variation in fish communities in the Hun River Basin. This is consistent with previous findings showing that fish community structure is strongly influenced by land use [21,24,53]. Wang et al. [54,55] found in the Wisconsin watershed in the United States that when the proportion of urban land in the basin reached 10–20%, the fish integrity index began to decrease significantly, while when the proportion of agricultural land exceeded 50%, the fish integrity index began to decrease significantly. Zhang et al. [56] studied the Taizi River and found that the taxic composition of fish changed significantly when the proportion of urban land was more than 2.6–3.1%. However, Ding et al. [24] also studied the Taizi River and found that when the proportion of agricultural land exceeded 25%, or the proportion of urban land exceeded 4%, significant changes occurred in the fish community structures. In this study, the sampling sites with less than 25% agricultural land and more than 60% forest land were dominated by sensitive species such as *Rhynchocypris lagowskii*, *Cobitis sibirica* and *Barbatula nuda*, while the sampling sites with more than 60% agricultural land and less than 5% forest land were dominated by tolerant species such as *Carassius auratus*, *Pseudorasbora parva* and *Hemiculter leucisculus*. These results are consistent with the results of Ding et al. [24], but they differ greatly from the results of Wang et al. [54,55]. This may be due to the fact that the Hun River and the Taizi River belong to the same first-level water ecological division and have similar climate, hydrology and habitat characteristics [57]; thus, they have similar response patterns to land use.

*4.3. Evaluation of Assessment Results Obtained Based on the F-IBI*

The fish biological integrity index reflects the degree of pressure caused by human disturbance on the river ecosystem. F-IBI evaluation results can guide water environment managers to make targeted restoration and protection measures, and so accuracy is expected by water environment managers [58]. The selection of reference sites and the screening of candidate indicators are the keys to ensure the accuracy of F-IBI evaluation results. Generally, the historical data of an evaluation water area is used or an original river section without human interference is selected as the reference point. At present, there is no unified selection standard [59]. As an urban river, the Hun River cannot be completely undisturbed by human activities, even at sites with high forest coverage. Therefore, according to the principle of being relatively unaffected and to the actual status of the Hun River Basin, six sites with good water quality (Class III), higher Shannon–Wiener indexes ($H' \geq 2$), fewer surrounding urban areas and higher vegetation coverage were selected as reference points, and they had higher levels of accuracy than relying solely on water quality or habitat quality ratings [60]. The candidate indexes covered five aspects, including species composition and abundance, nutrient structure, tolerance, reproductive co-location group, and habitat preference, which could fully reflect the fish community structure information, and the constructed F-IBI could scientifically evaluate the health status of the Hun River Basin. According to the calculation results of the F-IBI, two of the six reference sites were evaluated as healthy and four were evaluated as good, indicating that the selection of reference sites in this study was relatively appropriate.

*4.4. Implications for Protection and Rehabilitation*

Since few rivers are currently undisturbed by human activities [61,62], researchers are increasingly using extensive fish datasets to monitor the impacts of human activities on river ecosystems at the regional, national and global scales in order to provide sound recommendations for watershed governance and restoration [63,64]. The results of an F-IBI evaluation can help river basin managers to make targeted restoration and protection measures for different regions [65]. This study found that the F-IBI evaluation grades of the middle and upper reaches of the Hun River Basin were good and above good, while the F-IBI evaluation grades for the downstream regions were fair and below fair. The natural vegetation coverage rate in the middle and upper reaches was high and the population density was small, except for some agricultural production, and there was little industrial pollution and a low degree of human interference. In contrast, the lower reaches of the plains were densely populated, with densely populated towns, developed industries and agriculture, high levels of urbanization, and severe human disturbance. For the whole Hun River basin, the sampling points rated as fair grade or above fair accounted for 69.23% and the environmental pressure in the downstream area was greater. Although long-term ecological restoration has been ongoing since the implementation of the national water pollution control and treatment major scientific and technological projects, relevant measures have not been adjusted according to local conditions. In the present study, differences in spatial distribution among fish communities and environmental factors were found. The middle and upper reaches were relatively primitive and primarily contained sensitive fish. For this area, the current goal is to protect natural habitats. We recommend that priority should be given to the construction of protected areas, strengthening the protection of important fish habitats and enacting long-term fishing bans. In contrast, the downstream areas are currently severely disturbed by humans, and pollution-tolerant species are dominant, with fish communities being seriously degraded. For this region, ecological restoration is the current goal. Ecological restoration plays an important role in the restoration of aquatic ecosystems. Carrying out ecological restoration based on artificial restoration in the downstream areas where the ecosystem is seriously damaged can speed up the ecological restoration process and restore the water ecosystem. We suggest that priority should be given to improving the watershed management mechanism and system, updating fishery regulations and strengthening law enforcement, followed by

measures such as restoring regional habitats (converting farmland to forests and restoring riparian vegetation), strengthening water pollution control and ecological compensation, and building a long-term aquatic ecological monitoring system. After the water ecosystem has been initially restored, the improvement of the watershed management system can ensure that it can smoothly become a virtuous circle. The restoration of regional habitat not only improves the habitat environment of aquatic organisms, but it also indirectly improves the water quality. In addition, further strengthening water pollution control can more effectively improve water quality such that the water quality can meet the conditions for the growth of aquatic organisms, and at the same time, it can meet the needs of economic and social development and residents' lives.

## 5. Conclusions

This study investigated the fish community structures and environmental factors of the Hun River Basin in 2021. Subsequently, the CCA analysis method was used to determine the key environmental factors affecting the results of the fish communities. Finally, the F-IBI method was used to evaluate the aquatic ecological health. The results can be summarized as follows:

(1) A total of 51 species of fish were identified in the two seasons, belonging to 7 orders, 11 families and 37 genera. The dominant species of the fish community in the Hun River Basin were *R. lagowskii*, *Z. platypus*, *C. auratus* and *P. parva*. Fish communities were divided into two groups along the watershed longitudinal gradient. Agricultural land, urban land and grassland had significant effects on fish community.

(2) The health status of the water bodies in the Hun River Basin was mostly at the fair level or above. The F-IBI values were significantly different in spatial distribution, and the health status of the middle and upper reaches was better. The long-term ecological restoration had improved the health status of the downstream area, to some extent.

(3) According to the F-IBI assessment results, the middle and upper reaches should adopt natural habitat protection as a management goal, and the lower reaches should adopt ecological restoration as a primary goal.

**Author Contributions:** Conceptualization, J.X.; methodology, J.X. and C.W.; formal analysis, J.X.; investigation, J.X., C.W., L.L. and Y.D.; data curation, J.X.; writing—original draft preparation, J.X.; writing—review and editing, B.H. and D.L.; project administration, B.H.; funding acquisition, B.H. and D.L. All authors have read and agreed to the published version of the manuscript.

**Funding:** This study was supported by the Finance Special Fund of the Ministry of Agriculture and Rural Affairs—"Fisheries Resources and Environment Survey in the Key Water Areas of Northeast China".

**Institutional Review Board Statement:** Not applicable.

**Informed Consent Statement:** Not applicable.

**Data Availability Statement:** Not applicable.

**Acknowledgments:** The authors sincerely thank Rong Tang, Li Li, Xi Zhang and the students for their valuable suggestions in the experiment, and they thank Xuan Liu, Jiacheng She, Shiqi Gou and others for their help in field investigation.

**Conflicts of Interest:** The authors declare no conflict of interest.

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
