# Peer review of "Assessment of Aquatic Ecological Health Based on the Characteristics of the Fish Community Structures of the Hun River Basin, Northeastern China"

_water, doi:10.3390/w15030501_

Round 1

Reviewer 1 Report

This is a good paper. I suggest acceptance for publication based one some minor revisions.
1. The words "the Hunhe River" and "the Hun River" should be kept consisten in the full paper.
2. Line 19: All the Latin names should be in full species Latin names in the abstract.
3. Table 2: "Cyprinuscarpio" should be "Cyprinus carpio".

Author Response

Dear reviewer:

Thank you very much for your comments. We have revised the manuscript according to your suggestions. Please see the attachment.

Reviewer 2 Report

Dear authors and Editor I have reviewed the manuscript "Assessment of aquatic ecological health based on the characteristics of fish community structure of the Hunhe River Basin, northeastern China" submitted to WATER. The study investigated a very interesting question, i.e., the relationship between stream fish assemblages and environmental factors. The manuscript needs some improvements before acceptance for publication. My detailed comments as follow:
1. Line 128: authors used the Chinese national experimental standards, but the corresponding reference (No.33) do not correspond.
2. Line 142-145: authors should clarify the scale on which landscape data is measured, watershed scale or riparian scale. Besides, authors should clarify time period of the land use data.
3. Line 153-154: authors used the Shannon diversity index as a criteria for reference sites selection, but this criteria was just used in lake ecosystem. So the authors should elaborate on the applicability of this criteria in this study.
4. Line 198-204: for CCA analysis method, whether environmental variables include land use and water quality and water velocity? Because I just find land use types affecting fish assemblages from the result part.
5. Hierarchical cluster analysis (CA) results showed that all sampling points could be spatially divided into two groups (group A and group B). The results of Table3 compare the environmental factors between mid-upstream and downstream. I have doubts about the results of the two parts, whether the sampling points of the two parts are the same? If the sampling sites were different for the two parts, author should explain the division of upper, middle and lower region and the relative sampling site group.
6. The ecological management (water pollution control, water pollution treatment, afforestation, and seasonal fishing bans) play a positive role in river ecosystem recovery. For example, water pollution management, I think this work should be in the downstream areas, so how does that play a role in ecosystem restoration. I advise the authors to discuss it further.
7. Please check the fish scientific name in Table2, i.e., Cyprinuscarpio and Aristichthysnobilis.

Author Response

Dear Reviewer:

Thank you very much for your comments. We have responded to your questions and revised the manuscript based on your suggestions. Please see the attachment.
